# Sudden Death of a Four-Day-Old Newborn Due to Mitochondrial Trifunctional Protein/Long-Chain 3-Hydroxyacyl-CoA Dehydrogenase Deficiencies and a Systematic Literature Review of Early Deaths of Neonates with Fatty Acid Oxidation Disorders

**DOI:** 10.3390/ijns11010009

**Published:** 2025-01-26

**Authors:** Ana Drole Torkar, Ana Klinc, Ziga Iztok Remec, Branislava Rankovic, Klara Bartolj, Sara Bertok, Sara Colja, Vanja Cuk, Marusa Debeljak, Eva Kozjek, Barbka Repic Lampret, Matej Mlinaric, Tinka Mohar Hajnsek, Daša Perko, Katarina Stajer, Tine Tesovnik, Domen Trampuz, Blanka Ulaga, Jernej Kovac, Tadej Battelino, Mojca Zerjav Tansek, Urh Groselj

**Affiliations:** 1Department of Endocrinology, Diabetes and Metabolic Diseases, University Children’s Hospital, Ljubljana University Medical Center, Bohoriceva 20, 1000 Ljubljana, Slovenia; 2Faculty of Medicine, University of Ljubljana, Vrazov trg 2, 1000 Ljubljana, Slovenia; anaa.klinc@gmail.com (A.K.); jernej.kovac@kclj.si (J.K.); 3Clinical Institute for Special Laboratory Diagnostics, University Children’s Hospital, Ljubljana University Medical Center, Vrazov trg 1, 1000 Ljubljana, Slovenia; 4Institute of Pathology, Faculty of Medicine, University of Ljubljana, Korytkova 2, 1000 Ljubljana, Slovenia; 5Novo Mesto General Hospital, Smihelska cesta 1, 8000 Novo Mesto, Slovenia

**Keywords:** MTP deficiency, MTPD, LCHAD deficiency, LCHADD, fatty acid oxidation disorder, FAOD, sudden infant death, newborn, newborn screening, NBS

## Abstract

Mitochondrial trifunctional protein (MTP) and long-chain 3-hydroxyacyl-CoA dehydrogenase (LCHAD) deficiencies have been a part of the Slovenian newborn screening (NBS) program since 2018. We describe a case of early lethal presentation of MTPD/LCHADD in a term newborn. The girl was born after an uneventful pregnancy and delivery, and she was discharged home at the age of 3 days, appearing well. At the age of 4 days, she was found without signs of life. Resuscitation was not successful. The NBS test performed using tandem mass spectrometry (MS/MS) showed a positive screen for MTPD/LCHADD. Genetic analysis performed on a dried blood spot (DBS) sample identified two heterozygous variants in the *HADHA* gene: a nucleotide duplication introducing a premature termination codon (p.Arg205Ter) and a nucleotide substitution (p.Glu510Gln). Post-mortem studies showed massive macro-vesicular fat accumulation in the liver and, to a smaller extent, in the heart, consistent with MTPD/LCHADD. A neonatal acute cardiac presentation resulting in demise was suspected. We conducted a systematic literature review of early neonatal deaths within 14 days postpartum attributed to confirmed fatty acid oxidation disorders (FAODs), which are estimated to account for 5% of sudden infant deaths. We discuss the pitfalls of the NBS for MTPD/LCHADD.

## 1. Introduction

Mitochondrial trifunctional protein deficiency (MTPD) and long-chain 3-hydroxyacyl-CoA dehydrogenase deficiency (LCHADD; jointly abbreviated MTPD/LCHADD) are included in the population newborn screening (NBS) programs of many countries [1]. The condition has been part of Slovenia’s NBS since 2018 [2].

NBS for MTPD/LCHADD is based on establishing abnormal acylcarnitine profiles through tandem mass spectrometry (MS/MS), followed by genetic and enzymatic analysis to confirm the diagnosis. Confirmatory testing is necessary to reduce the risk of misdiagnosis of NBS [3,4] because a positive newborn screening result does not indicate a definitive diagnosis; it merely suggests a child at risk.

Acylcarnitine profiles do not fully discriminate between LCHAD, MTP, and long-chain ketoacyl-CoA thiolase (LCKAT) deficiencies. Thus, screening for LCHAD deficiency also results in screening for MTP and LCKAT deficiencies [5].

Inborn errors of metabolism (IEMs) are estimated to account for 0.9–6% of sudden unexpected death in infancy (SUDI), which refers to the death of a child that occurs suddenly and unexpectedly during the first year of life and represents one of the leading causes of post-neonatal death [6,7,8,9]. The IEMs associated with SUDI also include MTPD/LCHADD [5,10].

We describe a case of early lethal presentation of MTPD/LCHADD in a term newborn. In addition, we performed a systematic review of published cases of early neonatal mortality within the first 14 days postpartum due to fatty acid oxidation disorders (FAOD).

## 2. Case Report

The girl was born after an uneventful pregnancy and induced delivery at 40 weeks of gestation, with appropriate birth measures (BW 3.2 kg, BL 49 cm, HC 34 cm, Apgar 9/10). Meconium amniotic fluid was observed, but she was stable and did not need any interventions. She was discharged home at the age of 3 days and appeared well. At the age of 4 days, the newborn was fed as usual and at a regular interval, not showing any signs of fatigue or being unwell. After the girl awoke from a subsequent nap crying, the mother tried to comfort her before feeding, when the girl suddenly became unresponsive and showed no signs of life. No vital heart activity was achieved after 1 h of prolonged and adequate resuscitation. Asystole was the detected cardiac arrest rhythm and was unresponsive to treatment (11 × 12.5 mcg/kg of epinephrine, bicarbonate, glucose, and transfusion of concentrated erythrocytes and boluses with normal saline were administered). NBS samples were collected approximately 24 h before cardiac arrest.

The NBS test performed by MS/MS chronologically 3 days after death showed a positive screen for MTPD/LCHADD (reported in μmol/L with the maximum normal range in brackets): C14OH 0.45 (0.04), C16OH 2.77 (0.05), C16:1 OH 0.46 (0.1), C16 OH/C14 1.41 (0.2), C16 OH/C16 0.24 (0.03), C18:1 OH 1.17 (0.04), C18 OH 1.27 (0.03), and C18 OH/C18 0.93 (0.09). The screen report was available after the infant died.

Genetic next-generation whole exome sequencing analysis was performed after parental consent had been obtained, using DNA isolated from the dried blood spot (DBS) used for NBS. Two heterozygous pathogenic variants in the *HADHA* gene were found. The first was a nucleotide duplication c.612dup, resulting in a termination codon (p.Arg205Ter) that was previously reported as pathogenic (ClinVar ID 638987) and is pathologic according to the ACMG criteria (PVS1, PM2, PM3); the variant is not present in the healthy population (GnomAD database), and it was paternally inherited. The second variant, a nucleotide substitution 1528G>C (p.Glu510Gln), was reported as pathogenic (ClinVar ID: 100085, HGMD: CM 940884) and classified as pathogenic according to the ACMG criteria (PS3, PM2, PP3, PM3); the prevalence of the variant in the general population is 0.31% and it was maternally inherited. No other genetic variants were identified.

Post-mortem studies revealed a structurally normal heart with a persistent foramen ovale. Histology and electron microscopy findings were consistent with MTPD/LCHADD. Rather diffuse steatosis was present in the liver (Figure 1A,B), with small to medium intra-cytoplasmatic fat droplets in most hepatocytes. Fat droplets were also seen in fewer cardiomyocytes (Figure 1G,H). Additional electron microscopy (EM) (Figure 1C–F) revealed prominent cellular vacuolization with diminished organelles and only a few mitochondria showing deformed cristae structure. A neonatal acute cardiac presentation resulting in demise was suspected.

## 3. Materials and Methods of the Systematic Literature Review

We reviewed the literature on cases of the most common types of FAOD, in which death occurred within the first 14 days of life, referred to in this article as early neonatal death. A systematic literature review in the PubMed database was performed on 5 October 2024, following the PRISMA reporting guidelines (Appendix A). We searched the database for available case reports on patients with FAOD presenting as early neonatal death and collected data on pregnancy and delivery history, family medical history, initial clinical presentation and its onset, abnormalities in laboratory assessments, progression of symptoms, circumstances and timing of death, diagnostic approaches for FAOD, the timing and results of NBS, underlying genetic variants in cases in which FAOD was genetically confirmed, autopsy findings with a focus on histological results, and other post-mortem analyses. Search terms, including “neonatal death”, “sudden death”, “unexpected death”, “lethal”, “fatal”, and “SUDI”, were combined with full names and abbreviations of major types of FAOD listed in Table 1. We included case reports or case series documenting death within the first 14 days of life in patients diagnosed with FAOD via enzyme activity testing or genetic analysis, either pre- or post-mortem. In addition, we inspected references to the articles identified, and we identified some additional relevant reports. The references for the articles included and the number of patients presenting with early neonatal death for each major type of FAOD are detailed in Table 1. In certain types of FAOD, the actual number of early neonatal deaths may be higher than reported because some case reports describe a sibling that died under very similar circumstances without FAOD being diagnosed at the time; however, the condition was later confirmed in a younger sibling.

## 4. Results of the Systematic Literature Review

### 4.1. Medium-Chain Acyl-CoA Dehydrogenase Deficiency (MCADD)

In most cases, the pregnancy and delivery were uneventful, with symptoms emerging between 12 h and 5 days, including difficulty feeding, lethargy, and hypotonia. Vomiting and seizures can occur. There are more than 19 cases of MCADD patients with early neonatal death described in the literature. Profound hypoketotic hypoglycemia was described in all instances in which glucose measurement during the metabolic decompensation was reported. Frequently, lactic acidosis and hyperammonemia occur. Death occurred due to sudden cardiac-respiratory arrest or multi-organ failure at a median age of 3 days (range: 44 to 144 h) [11,12,13,14,15,16,17,18,19,20,21,22]. Common post-mortem findings included extensive hepatic steatosis and fatty vacuolization of cardiomyocytes, renal tubular epithelium, and skeletal muscle fibers. Almost all cases described were homozygous for the pathogenic *ACADM* variant c.985A>G (p.Lys329Glu) [11,12,13,14,15,16,17,18,19,20,21,22].

### 4.2. Mitochondrial Trifunctional Protein/Long-Chain 3-Hydroxyacyl-CoA Dehydrogenase Deficiencies (MTPD/LCHADD)

The initial presentation occurred within the timeframe of 12 h to 8 days. The main characteristics described in the literature for 32 patients with MTPD/LCHADD with lethal neonatal phenotype were severe cardiomyopathy and lactic acidosis. Fetal abnormalities such as biventricular cardiac hypertrophy and *hydrops fetalis* were observed in utero in some cases. In all patients with available laboratory reports, there was metabolic acidosis with limited response to treatment; hypoketotic hypoglycemia, elevated creatine kinase (CK), and mild hyperammonemia were somewhat less frequently present. The clinical condition rapidly deteriorated due to cardiomyopathy at a median age of 5 days (range: 0.5 to 14 days). Hypotonia and liver involvement can be present [27,28,29,30,31,32,33,34,35,36,37,38].

The predominant post-mortem finding was dilatative or hypertrophic cardiomyopathy; fatty infiltration of the liver, heart, and skeletal muscle was generally moderate to mild and not universally present. In cases in which NBS could detect MTPD, patients were severely ill or deceased before the results became available. Enzyme activity determinations in cell cultures revealed pronounced deficiencies in LCHAD and LKAT activities, consistent with MTPD characteristics. According to genetic analysis, the disease-causing variants are more common in the *HADHB* gene than in the *HADHA* gene [27,28,29,30,31,32,33,34,35,36,37,38], with the 1528G>C variant being the most prevalent [91,92,93,94].

The occurrence of early neonatal death in isolated LCHADD is less frequent than in complete MTPD. At the median age of three days (range: 2 to 7 days), patients experienced sudden and unexpected death, with minimum or no preceding symptoms. Severe hypoketotic hypoglycemia was characteristic. The only reported autopsy finding was hepatic steatosis [39,40,41,42,43]. Even in cases with a prevented fatal outcome, numerous MTPD/LCHADD patients presented with symptoms in the days before the NBS results were available [26,43,44,45]. Sykut-Cegielska et al. described a case of early neonatal death due to LCHADD at the age of 7 days; the diagnosis was confirmed from the NBS DBS sample the same day [26].

### 4.3. Very Long-Chain Acyl-CoA Dehydrogenase Deficiency (VLCADD)

Over nine cases of VLCADD patients with a fatal neonatal outcome have been reported, with death occurring shortly after birth. Sudden cardiac death occurred at a mean age of 43 h (range: 1 to 7 days); most cases were asymptomatic before that. In rare instances, grunting, hiccups, and arrhythmias were noticed. Liver disease, cardiomyopathy, and pericardial effusion can be present. Several cases describe hypoketotic hypoglycemia, metabolic acidosis, and moderately elevated liver enzymes. Feeding difficulty was noted in a single case. Resuscitation efforts proved unsuccessful upon finding an unresponsive child [46,47,48,49,50,51,52]. The significant autopsy findings included microvesicular steatosis of the liver and fatty degeneration of the myocardium. Most patients died before the NBS results were available or even before screening samples were taken. A diverse array of variants were detected in the *ACADVL* gene, with no single variant standing out in frequency [46,47,48,49,50,51,52].

### 4.4. Carnitine Uptake Defect (CUD)

The patient reported by Rinaldo et al. is the only case of early neonatal death due to CUD. Lethargy and feeding difficulty were observed from the 1st day until the infant was found without signs of life on day 5. Post-mortem analysis revealed hypoglycemia and elevated levels of C10–C18 fatty acids, coupled with microvesicular fatty infiltration of the liver and myocardium. CUD was confirmed through a significantly reduced total carnitine concentration and carnitine transport assay in the parents’ cultured fibroblasts [53]. CUD is rare in the European population, except in the Faroe Islands and Denmark, where the c.95A>G variant is most prevalent in severe cases [95,96].

### 4.5. Carnitine Palmitoyltransferase I Deficiency (CPT1D)

One neonatal patient with a fatal outcome due to CPT1D was reported by Invernizzi et al. Death occurred after an episode of treatment-resistant bradycardia 34 h after birth. The autopsy revealed microvesicular steatosis of the liver, with no abnormalities detected in the heart. The NBS sample revealed an elevation of free carnitine, a reduction in acetylcarnitine, and a near absence of other acylcarnitines. A profound decrease in CPT1 activity in cultured fibroblasts confirmed CPT1D. Presumably, CPT1D was also the cause of death in his sister, who passed away under similar circumstances at 3 days of age [54]. CPT1D is rare in Europe, with no specific high-frequency variants, except for individuals of Inuit origin, where the c.1436C>T variant in *CPT1A* is the most prevalent [97].

### 4.6. Carnitine Palmitoyltransferase II Deficiency (CPT2D)

Severe neonatal phenotype is one of the typical presentations of CPT2D. We reviewed 18 cases in which the initial presentation typically occurred between 12 h and 12 days postpartum, resulting in early neonatal death. Compared to other FAODs, CPT2D is characterized by cystic dysplasia of the renal parenchyma and renal insufficiency. Initial symptoms include respiratory distress, hypotonia, lethargy, and hypothermia. The clinical course was complicated by resistant seizures and episodes of apnea requiring mechanical ventilation. Bradycardia and widening of the QRS complexes are the leading causes of death. The median time of death is the 5th day after birth (range: 1.5 to 14 days); it is slightly later compared to other FAODs because approximately half of the reported patients die in the 2nd week of life [55,56,57,58,59,60,61,62,63,64,65,66,67,68].

In addition to nephrological changes, the autopsy findings include cardiomegaly with biventricular hypertrophy, hepatomegaly, and intracytoplasmic lipid accumulation in hepatocytes, skeletal and myocardial muscle fibers, renal epithelial tubules, and the adrenal cortex. In some cases, neurodevelopmental anomalies were also reported. Hypoglycemia and liver calcifications can also be present. In nearly all cases, the disease was also genetically confirmed; no detected variant in the *CPT2* gene predominates [55,56,57,58,59,60,61,62,63,64,65,66,67,68].

### 4.7. Carnitine-Acylcarnitine Translocase Deficiency (CACTD)

Symptoms of CACTD commonly manifest in the neonatal period [69]; hence, descriptions of early neonatal death are prevalent. The disease manifests in the timeframe between 0.5 and 52 h postpartum, with most patients experiencing onset on the 1st or 2nd day of life. In the literature, fatal neonatal outcomes have been reported in more than 32 patients with diagnosed CACTD, with additional cases in which the diagnosis was established in a sibling. Typically progressing rapidly, the disease is mainly characterized by the abrupt onset of cardiorespiratory insufficiency. Patients are often found in sudden cardiac arrest, lacking prior symptoms. In laboratory assessment, hypoglycemia, hyperammonemia, and lactic acidosis are common, and elevations in CK, transaminases, and lactate dehydrogenase (LDH) are often associated. Episodic bradycardia with hypotension and recurrent ventricular tachycardia leading to ventricular fibrillation were most frequently recorded; patients died at a median age of 3 days (range: 1 to 9 days) due to cardiac-respiratory arrest. Even rigorous diet and carnitine supplementation interventions could not prevent a fatal outcome [61,69,70,71,72,73,74,75,76,77,78,79,80,81,82,83,84,85,86,87,88,89,90]. On autopsy, severe lipid cardiomyopathy, hepatomegaly, and diffuse fatty liver and kidney infiltration were described. In the severe neonatal phenotype, cultured fibroblasts usually have no detectable CACT enzyme activity. Various disease-causing variants in the *SLC25A20* gene were detected, with c.199-10T>G splice site change being particularly common [61,69,70,71,72,73,74,75,76,77,78,79,80,81,82,83,84,85,86,87,88,89,90].

## 5. Discussion

Although the clinical presentations vary, FAODs are an important group of IEMs, accounting for high morbidity and mortality. Premature death is common in FAODs, and they are estimated to cause 5% of sudden and unexpected deaths in infants [5,10]. As shown in our systematic review, the most common FAODs causing early neonatal death are CACT and MTPD/LCHADD.

The only non-conclusive FAOD is SCADD because the metabolic findings might just be incidental in these circumstances; SCADD is now viewed as a biochemical phenotype rather than a disease and should not preclude additional testing to look for other causes of the clinical picture [98]. SCADD is being removed from the NBS panels [99]. These are the reasons why we did not include this condition in our review.

The timing of presentations ranges from under 24 h to 14 days, with our described case fitting into the calculated median of published MTPD/LCHADD cases. Even beyond fatal neonatal presentation, long-term complications of MTPD/LCHADD include severe liver disease, peripheral neuropathy, and retinopathy [44,100]. Approximately 38% of infants die before or within 3 months of MTPD/LCHADD diagnosis [101].

In patients with MTPD/LCHADD, a high incidence of prematurity (>60%) and maternal hemolysis, and elevated liver enzymes, low platelet count (HELLP) syndrome (approximately 25%) is found, which adds to the disease burden [102].

The genotype of the case presented is compound heterozygosity for the 1528G>C variant, which is a high-frequency mutation present in European patients with clinically overt disease in homozygous or heterozygous form with a second *HADHA* polymorphism, as it was in our case [91,92,93,94]. Determining enzymatic activity is the only way to characterize isolated LCHAD deficiency or general MTP deficiency; however, this was not possible in our case. The clinical presentation of our case does not differ from other published cases.

Several management strategies are available; however, there is no cure for MTPD/LCHADD [103]. This pitfall covers NBS for all FAOD. MTPD/LCHADD leads to an accumulation of toxic β-oxidation intermediates, causing acute and long-term symptoms not responding to available treatment protocols [10,26]. Even patients diagnosed with MTPD/LCHADD prenatally and treated rigorously with intravenous glucose, sodium bicarbonate, diet, and carnitine supplementation died due to cardiac decompensation [61,62,69,70,71,72,73,74,75,76,78,79,80,81,82,83,84,85,86,87,88,89]. Comparing the health outcomes of people with MTPD/LCHADD treated with pre-symptomatic dietary management following NBS detection with people detected following symptomatic presentation found no statistically significant differences in outcomes between the two groups [104].

However, treated in time, the outcome of MTPD/LCHADD can be favorable, prompting its inclusion into NBS programs [44,95,103,105,106].

Since 2018, the Slovenian NBS program has included a group of FAODs using tandem mass spectrometry (MS/MS) and confirmatory whole genome sequencing (WES) testing [2], with VLCADD (eight cases) and MCADD (eight cases) being the most prevalent [107]. The cumulative incidence of FAOD in the NBS era in Slovenia (18.7 per 100,000 newborns screened in the period 2018–2023) is slightly higher compared to the general incidence of 0.9 to 15.2 per 100,000 [108].

There is currently insufficient evidence to judge the test accuracy of acylcarnitine profiling from DBS for MTPD/LCHADD. The NBS programs do not use the same combination of markers and thresholds for the FAOD [103]. A systematic review performed by Stilton et al. in 2021 showed that positive predictive value in the 10 studies included ranged from 0% (zero true positives and 28 false positives from 276,565 babies screened) to 100% (13 true positives from 2,037,824 babies screened). No sensitivity, specificity, or negative predictive value could be calculated because there was no systematic follow-up of babies that had screened negative [103].

For some FAODs, missed cases of NBS were reported, most commonly for VLCADD. This is due to possibly normal or only slightly elevated VLCADD-specific biomarkers in NBS tests [109,110]. Furthermore, sporadic cases missed in the initial NBS sample have been reported for MTPD/LCHADD. Lotz-Havla et al. described three cases resulting in life-threatening metabolic decompensations within the first 6 months of life. One was overlooked due to misinterpretation caused by prematurity, and the other two were missed due to inappropriate management of confirmatory testing [5].

Modifying analyte cut-off values, implementing analyte ratios, second-screening specimens in preterm babies, or second-tier strategies, including next-generation sequencing [3,4], can be implemented to avoid false negative results [111,112].

For MTPD/LCHADD, it has been reported that immediate metabolic advice is crucial for the survival of newborns detected by newborn screening. Most deaths occur before or within 3 weeks of diagnosis, especially in patients with a limited approach to metabolic expertise [26].

In Slovenia, all newborns that test positive for NBS are actively invited for further testing by the Metabolic Center of the Ljubljana Children’s Hospital, so that appropriate confirmatory testing can be performed and lost to follow-up cases are reduced. When a newborn is admitted to the neonatal intensive care in a peripheral maternity hospital, a consulting metabolic specialist gives advice on therapy and diet regime, plans the actions to be taken, and discusses the screening results with the parents.

Our literature review shows that a recognizable latent or early symptomatic stage does not exist in some FAOD cases. Therefore, the fatal outcome of FAOD could not be prevented by NBS in all cases. For example, for MCADD, there are reports that the NBS results were unavailable before the onset of acute clinical events [14,15,16,21,113]. Furthermore, Estrella et al. reported an MCADD patient that died before the collection of the NBS sample [111]. Nevertheless, NBS has significantly contributed to reducing the overall neonatal mortality rate in MCADD (0.6% in Germany and 2.4% in Australia) in comparison to the period before the inclusion of MCADD in NBS (7.5% in an Australian unscreened population) [15,114], and positive outcomes are also convincing in the Slovenian NBS program [115]. Numerous MTPD/LCHADD patients presented with symptoms in the days before the NBS results were available [26,43,44,45]. The results of NBS were often available too late in cases of CACTD [69,70,71,72,73,74,75,76,78,79,80,81,82,83,84,85,86,87,88,89].

There must be no delays in processing the NBS samples—from adherence to the agreed time of sample taking to sending, analyzing, interpreting, and communicating the results promptly [116]. In our case report, we describe a newborn that died 24 h after the NBS sample collection and could not be helped. However, determining the reason for the sudden death was valuable to the parents and medical personnel. The parents received genetic counseling to plan further pregnancy and also avoid the risk of the mother developing HELLP syndrome because a distinctive feature of this type of FAOD is the frequent occurrence of pregnancy complications (hemolysis, elevated liver enzymes, low platelet count, pre-eclampsia, and acute fatty liver) in the mother carrying an affected child; this was reported in approximately a third of MTPD/LCHADD cases resulting in early neonatal death [15,16,17,23,24,25,27,28,29,30,31,32,33,34,35,36].

Screening for MTPD/LCHADD is conducted as part of the NBS programs in several countries, but more published data on the benefits and harms of the programs must be collected. The elements that might determine the balance of benefits and harms from screening programs are test accuracy, the benefit of early detection and treatment, and overdiagnosis—that is, detection and treatment of a condition that never would have caused symptoms within a person’s lifetime [104,117]. There are no published data on that aspect. The lack of data and variability between studies leads to considerable uncertainty regarding the benefits and harms of screening for MTPD/LCHADD [103,104].

False-positive screening results do create a high burden for the NBS programs but even more so for families, with an impact on parental anxiety, stress, and possibly altered parent–child relationships. However, the experience of a life-threatening decompensation due to an insufficiently diagnosed metabolic disorder severely traumatizes parents and advocates in favor of NBS [5,118,119].

## 6. Conclusions

Our observations highlight the main challenges of NBS for MTPD/LCHADD. Adherence to the time frame for collecting, sending, and analyzing the samples is essential. Rapid action is needed in the case of an initial positive NBS result, which should be communicated to the attending physician and the parents by a medical professional in charge of treating MTPD/LCHADD. The results may not be reported soon enough to identify severely affected infants.

Rapid confirmatory genetic testing should be available for the FAODs in the NBS programs. It is important for false positive cases in the sense of shortening the parental stress of coping with a pending result, and even more so for newly diagnosed patients. It is also important in the case of SUID to identify the cause of death and guide genetic counseling for the family.

Clinicians caring for MTPD/LCHADD patients may consider partnership development across clinical and research networks to address the pitfalls of the NBS programs for FAODs, to produce per-country disorder-based overviews to compare screening performance, diagnostic confirmation, crisis interventions [120], and therapy follow-up for FAODs, and to advocate for advancements and work toward achieving equity in the NBS programs worldwide [116,121,122]. As a next step, we would consider surveying centers involved in the NBS for FAOD to obtain further information on challenges and existing best practices.

## Figures and Tables

**Figure 1 IJNS-11-00009-f001:**
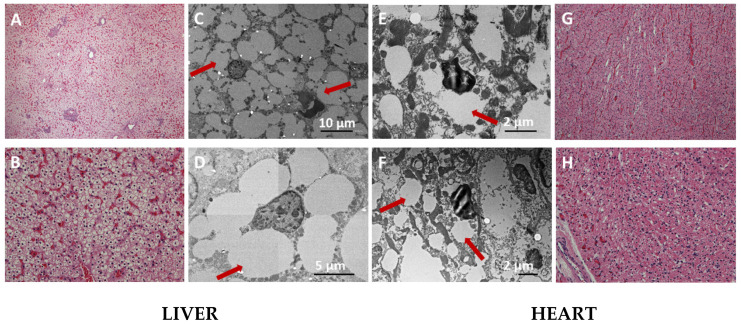
Post-mortem histology tests: diffuse accumulation of fat droplets within the hepatocyte (**A**–**D**) and cardiomyocyte (**E**–**H**) sections. Arrows show fat accumulation detected on EM (**C**–**F**). (**A**) Liver, HE stain 4× magnification, the orthotopic liver structure is preserved. There is diffuse steatosis, and optically clear vacuoles are present in nearly all hepatocytes. (**B**) Liver, HE stain 20× magnification, diffuse steatosis. Sinusoids are congested. (**C**) Liver on EM. (**D**) Single hepatocyte on EM. (**E**) Cardiac tissue on EM. (**F**) Cardiocyte on EM. (**G**) Myocard, HE stain 4× magnification, the orthotopic structure is preserved. Intracytoplasmatic clear vacuoles (fat droplets) are present within cardiomyocytes. (**H**) Myocard, HE stain, 20× magnification. Legend: HE stain hematoxylin and eosin stain, M magnification, EM electron microscopy.

**Table 1 IJNS-11-00009-t001:** Data from a systematic literature review of fatal FAOD cases in the first 14 days of life.

FAOD Type	NBS Results at Time of Presentation	No. of Cases Described	Median Age at Presentation (Range)	Median Age at Death (Range)	References
MCADD	R not available	14	48 h (12–120 h)	3 d (2–6 d)	[11,12,13,14,15,16,17,18]
	No NBS	>5	46 h (24–70 h)	3 d (2.5–4 d)	[12,19,20,21,22]
MTPD/LCHADD	R not available	5	2 d (1–3 d)	5 d (3–10 d)	[23,24,25]
	R available	1	7 d	7 d	[26]
	No NBS	>26	3 d (0.5–13 d)	5 d (0.5–14 d)	[26,27,28,29,30,31,32,33,34,35,36,37,38,39,40,41,42,43,44,45]
VLCADD	R not available	4	30 h (24–40 h)	38 h (32–48 h)	[46,47,48,49]
	No NBS	>5	41 h (1–3 d)	2 d (1–3 d)	[50,51,52]
CUD	R not available	—			
	No NBS	1	1 d	5 d	[53]
CPT1D	R not available	—			
	No NBS	1	34 h	34 h	[54]
CPT2D	R not available	6	20 h (17–240 h)	3 d (1.5–13 d)	[55,56,57,58,59,60,61]
	R available	2	10 d	13.5 d (13–14 d)	[55,60]
	No NBS	10	36 h (12–72 h)	5 d (1.5–12 d)	[57,62,63,64,65,66,67,68]
CACT	R not available	>20	24 h (0.5–52 h)	3 d (1.5–8 d)	[61,69,70,71,72,73,74,75,76,77]
	No NBS	>12	30 h (24–48 h)	1.5 d (1–10 d)	[78,79,80,81,82,83,84,85,86,87,88,89,90]

NBS—newborn screening; R—results; h—hours; d—days; MCADD—medium-chain acyl-CoA dehydrogenase deficiency; MTPD/LCHADD—mitochondrial trifunctional protein/long-chain 3-hydroxyacyl-CoA dehydrogenase deficiencies; VLCADD—very long-chain acyl-CoA dehydrogenase deficiency; CUD—carnitine uptake defect; CPT1D—carnitine palmitoyltransferase I deficiency; CPT2D—carnitine palmitoyltransferase II deficiency; CACT—carnitine-acylcarnitine translocase deficiency.

## Data Availability

Data are available upon request.

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
