# Peer review of "Sudden Death of a Four-Day-Old Newborn Due to Mitochondrial Trifunctional Protein/Long-Chain 3-Hydroxyacyl-CoA Dehydrogenase Deficiencies and a Systematic Literature Review of Early Deaths of Neonates with Fatty Acid Oxidation Disorders"

_2409-515X, 2025, doi:10.3390/ijns11010009_

Round 1

Reviewer 1 Report

Comments and Suggestions for Authors

In this manuscript, the authors present a case of death at 4 days of age in an infant ultimately shown to have TFP/LCHAD deficiency. FAODs in general are a well know cause of infant deaths in non-screened babies, and have been reported even in infants screened at birth due to delayed receipt of NBS results. Thus, the authors perform a literature review of newborn deaths due to FAODs and present their results. Their methods are sound and results unsurprising, but serve as a nice reminder that NBS does not solve the problem of early newborn death unless the results are available before symptoms. Fortunately, these cases are still the exception and not the rule. 

I have some specific comments to improve the manuscript:

Line 42: acylcarnitien should not be capitalized

Line 64: Suggest using "mother" instead of "mum"

Line 70: Clarify the the NBS test was performed 3 days after death on a sample obtained prior to discharge. In the discussion, I would then emphasize the need to perform such tests expediently. Doing an NBS 7 days after birth is unacceptable laboratory procedure. 

Line 97: Please confirm the starting date for the search criteria. 

LIne 121: In the table, I suggest differentiating between cases with early onset of disease before return of NBS results vs those who predated NBS screening. 

LIne 229: In the results, I suggest more emphasis on NBS timing and do so at the beginning of the results. 

LIne 271: should not be indented as it is a continuation of a previous paragraph

Author Response

Response to Reviewer 1

Thank you for reviewing this manuscript and for your thoughtful and constructive comments. Please find the detailed responses below and the corresponding revisions/corrections highlighted/in track changes in the re-submitted file.

Comment 1: Line 42: acylcarnitien should not be capitalized

Response 1: Thank you for pointing this out; the change was made.

Comment 2: Line 64: Suggest using "mother" instead of "mum"

Response 2: We agree with the suggested change.

Comment 3: Line 70: Clarify the  NBS test was performed 3 days after death on a sample obtained prior to discharge. In the discussion, I would then emphasize the need to perform such tests expediently. Doing an NBS 7 days after birth is unacceptable laboratory procedure.

Response 3: We confirm the data and agree with your suggestion.

The NBS test was performed 3 days after death (and 7 days after birth), while the laboratory analysis was performed on the same day the sample arrived at the NBS laboratory. We could not find out what exactly caused the delay in sending the sample. Maternal hospitals are legally obliged to take the sample 48-72 hours after birth and send it in 24 hours after it is taken. We briefed our concerns to all the Maternity hospitals, and it is officially advised that samples are sent by a carrier, not by regular mail; however, unwanted delays still happen occasionally, mostly in cases where regular mail service is used. One of the aims of this article was to re-emphasize the importance of prompt sample analysis; we do not want to blame Maternal Hospitals for delays but to make sure everyone in the process understands the importance of adherence to protocols in NBS. Thus, we could not agree more with your opinion of those delays being unacceptable and should be prevented.

The text was added to the discussion in lines 309-311: There must be no delays in processing the NBS samples- from adherence to the agreed time of sample taking to sending, analyzing, interpreting and communicating the results promptly [117].

Comment 4: Line 97: Please confirm the starting date for the search criteria.

Response 4:  We confirm the date.

The Literature search was performed on October 5th, 2024.

Comment 5: Line 121: In the table, I suggest differentiating between cases with early onset of disease before return of NBS results vs those who predated NBS screening.

Response 5: Thank you for pointing this out; we agree this distinction is important. Therefore, we added data in the 2nd column of Table 1.

Comment 6: Line 229: In the results, I suggest more emphasis on NBS timing and do so at the beginning of the results.

Response 6:  We changed Table 1 in Results to include the relation to NBS. However, we were not able to find any further information on timing as were already presented.

Comment 7: Line 271: should not be indented as it is a continuation of a previous paragraph.

Response 7: Thank you for noticing the mistake; we corrected the paragraph layout.

Reviewer 2 Report

Comments and Suggestions for Authors

This article is a case report of sudden neonatal death from TFP deficiency experienced in Slovenia. However, the boundary between a case report article and a review article is unclear. It is unfortunate that the main point of this paper is unclear.

It is already known that LCHAD/TFP deficiency and CACT deficiency are FAODs that most  often cause sudden death in the neonatal period, although the other diseases could also cause neonatal deaths, in smaller part.

 If this paper is a case report of sudden neonatal death with TFP deficiency, the characteristics of this case should be more clearly described. The results of newborn screening in Slovenia, the frequency of each FAOD and the comparison of LCHADD/TFPD frequencies and genotypes in the world and in Slovenia should be discussed. Genotypes including high frequency variants of the HADA gene in Europe, and defective protein (LCHAD, LCEH, or LCKAT) should be discussed.

Although the review of case reports of FAOD that caused the neonatal death put in the results section, it only picks up on case reports and seems to have little novelty. Also, this part should be included in Discussion section.

Author Response

Response to Reviewer 2

Thank you very much for taking the time to review this manuscript. Please find our responses below and the corresponding revisions/corrections highlighted/in track changes in the re-submitted file.

Comment 1: This article is a case report of sudden neonatal death from TFP deficiency experienced in Slovenia. However, the boundary between a case report article and a review article is unclear. It is unfortunate that the main point of this paper is unclear.

Response 1: Thank you for pointing this out. We better delineated both parts of the manuscript to improve its clarity.  We aimed to go beyond only case presentation by adding a systematic review on the related topic, as it is a well-established way of presentation when reporting on rare disease cases.

Specifically, our aim was to collect similar cases of FAOD-related early neonatal deaths in the era of NBS.

We believe readers would appreciate the comprehensive presentation of FAOD-related early neonatal deaths, since this could be of great importance for application of newborn screening programs for FAOD.

We emphasized the distinction in the new subtitle order.

Comment 2: If this paper is a case report of sudden neonatal death with TFP deficiency, the characteristics of this case should be more clearly described.

Response 2: We included all the data available. However, the clinical data were very limited as the newborn was discharged as a presumably healthy child, after uneventful pregnancy, birth and postpartum period, only to be unexpectedly found in cardiac arrest the following day at home. The medical professionals later encountered the newborn girl only in a reanimation setting. No other clinical data was available. All the applicable post-mortem findings were included.

Comment 3: The results of newborn screening in Slovenia, the frequency of each FAOD and the comparison of LCHADD/TFPD frequencies and genotypes in the world and in Slovenia should be discussed. Genotypes including high frequency variants of the HADA gene in Europe, and defective protein (LCHAD, LCEH, or LCKAT) should be discussed.

Response 3:  Thank you for this observation. We added the data available.

The Slovenian NBS program has included FAOD since 2018. In those years, 8 MCADD, 8 VLCADD and this described LCHADD case were diagnosed (18.7 per 100 000 newborns screened for the 2018-2023 period), which is slightly higher compared to the general incidence of 0.9 to 15.2 per 100 000. We have a historic cohort before the NBS of four patients, two of them died in the first year of life.

The text now reads in lines 266-270: Since 2018, the Slovenian NBS program has included a group of FAODs using MS-MS and confirmatory WES testing [2], with VLCADD (8 cases) and MCADD (8 cases)  being the most prevalent [103]. The cumulative incidence of FAOD in the NBS era in Slovenia (18.7 per 100,000 newborns screened for the 2018-2023 period) is slightly higher than the general incidence of 0.9 to 15.2 per 100,000 [118].

And in lines 77-85 additional data was added:

Two heterozygous pathogenic variants in the HADHA gene were found. The first is nucleotide duplication NM_000182.5:c.612dup, resulting in a termination codon (p.Arg205Ter) that was previously reported as pathogenic (ClinVar ID 638987) and is pathologic according to ACMG criteria (PVS1, PM2, PM3); the variant is not present in the healthy population (GnomAD database) and was paternally inherited. The second variant, a nucleotide substitution NM_000182.5 .1528G>C (p.Glu510Gln), was reported as pathogenic (ClinVar ID: 100085, HGMD: CM 940884) and classified as pathogenic according to ACMG criteria (PS3, PM2, PP3, PM3), the prevalence of the variant in general population is 0.13% and was maternally inherited. ]

Comment 4: Although the review of case reports of FAOD that caused the neonatal death put in the results section, it only picks up on case reports and seems to have little novelty. Also, this part should be included in Discussion section.

Response 4: We aimed to perform the systematic literature review in the results section, in a well established scientific way with followingthe PRISMA methodology for systematic literature reviews. Usually with systematic reviews the results are presented under the Results section, and are discussed further in the Discussion section. We believe that it is very important to aggregate the individual case reports in such a systematic way to allow for stronger conclusions, especially in cases of ultra-rare disorders, where it is very usual to encounter individual case reports that provide limited insight.

Round 2

Reviewer 2 Report

Comments and Suggestions for Authors

This paper is a case report of TFP deficiency. In addition, an overview of FAOD is reviewed for each disease. In this paper, however, the case presentation of TFP deficiency and the systemic review of FAODs are seemed to be independent.

The clinical, pathological, and molecular biological findings of this case should be discussed more, comparing with the literature review of TFP deficiency. For example, the 1528 variation in LCHAD domain is famous as a high frequency in European population. It should be mentioned compared with non-European population in the review.

 In other diseases, the 985 mutation in MCADD is notorious for its high frequency in European populations and is described in the MCADD section; if there are high frequency mutations in other FAODs, including TFP deficiency, this should also be mentioned. 

 Genetic, pathologic, and clinical features of FAODs, including TFP deficiency, could be added shortly in the reviews. Re-review is not required.

Author Response

Thank you so much for new comments.

Please find our response and additions in the file attached.

Kind Regards,

Ana Drole Torkar
